# Gaussian Head & Shoulders: High Fidelity Neural Upper Body Avatars with Anchor Gaussian Guided Texture Warping

**Tianhao Wu**
University of Cambridge

**Jing Yang**
University of Cambridge

**Zhilin Guo**
University of Cambridge

**Jingyi Wan**
University of Cambridge

**Fangcheng Zhong**
University of Cambridge

**Cengiz Oztireli**
University of Cambridge

## Abstract

The ability to reconstruct realistic and controllable upper body avatars from casual monocular videos is critical for various applications in communication and entertainment. By equipping the most recent 3D Gaussian Splatting representation with head 3D morphable models (3DMM), existing methods manage to create head avatars with high fidelity. However, most existing methods only reconstruct a head without the body, substantially limiting their application scenarios. We found that naively applying Gaussians to model the clothed chest and shoulders tends to result in blurry reconstruction and noisy floaters under novel poses. This is because of the fundamental limitation of Gaussians and point clouds – each Gaussian or point can only have a single directional radiance without spatial variance, therefore an unnecessarily large number of them is required to represent complicated spatially varying texture, even for simple geometry. In contrast, we propose to model the body part with a neural texture that consists of coarse and pose-dependent fine colors. To properly render the body texture for each view and pose without accurate geometry nor UV mapping, we optimize another sparse set of Gaussians as anchors that constrain the neural warping field that maps image plane coordinates to the texture space. We demonstrate that Gaussian Head & Shoulders can fit the high-frequency details on the clothed upper body with high fidelity and potentially improve the accuracy and fidelity of the head region. We evaluate our method with casual phone-captured and internet videos and show our method archives superior reconstruction quality and robustness in both self and cross reenactment tasks. To fully utilize the efficient rendering speed of Gaussian splatting, we additionally propose an accelerated inference method of our trained model without Multi-Layer Perceptron (MLP) queries and reach a stable rendering speed of around 130 FPS for any subjects.

## 1 Introduction

Personalized and controllable 3D head avatar is a crucial asset for interactive Mixed Reality and metaverse applications. Recent developments in the 3D representations such as 3DMM (Li et al., 2017; Gerig et al., 2017), Neural Radiance Field (Mildenhall et al., 2020), Instant Neural Primitives (Müller et al., 2022), and other implicit representations (Mescheder et al., 2019) have brought rapid advancements in the reconstruction of vivid and controllable neural avatars (Zheng et al., 2022; Grassal et al., 2021; Zielonka et al., 2022; Gao et al., 2022). With the most recent 3D Gaussian Splatting representation (Kerbl et al., 2023), neural avatars can be convincingly reconstructed from a monocular video with impressive fidelity. However, most current methods for creating head avatars concentrate solely on the face and head, discarding other visible parts of the body by using a semantic mask during the training process. Consequently, this results in avatars that appear as heads without bodies, which is not sufficient for many immersive applications, including video conferencing, where a more complete avatar is needed (Shao et al., 2024; Xiang et al., 2024; Zielonka et al., 2022; Gao et al., 2022). Recent techniques aim to create more complete avatars by including visible

parts of the body, like shoulders and chest (Zheng et al., 2023; Zhao et al., 2024; Wang et al., 2024; Zheng et al., 2022). However, they are limited to simplified settings where the subject dresses in plain clothing without detailed textures and is instructed to restrict upper body movement. On the other hand, existing full-body avatar methods typically focus on the overall quality of the limbs and torso and fail to faithfully capture the fine details such as high-frequency texture on clothes (Kocabas et al., 2023; Hu et al., 2024b; Li et al., 2024; Lei et al., 2023). Applications of neural avatars that require detailed reconstruction of the upper body area often encounter significant challenges in capturing faithful and intricate details. Overall, current methods still fall short of delivering the level of performance needed for practical, real-world use.

The Gaussian Splatting representation, while being efficient and effective in certain aspects, struggles with accurate modeling of clothed upper bodies. As one of its fundamental limitations, each Gaussian can represent only one color from a specific viewing angle. This heavily limits its capability to handle dynamic objects that have complex textures, such as clothing with intricate patterns. To capture the detailed appearance of such objects, an excessively large number of Gaussians would be needed, increasing memory requirement and slowing down the rendering speed. In addition, the complicated pose-dependent appearances such as brightness changes and cloth wrinkles further increase the difficulty of modeling them with plain Gaussians alone. As a result, when the reconstructed avatar is driven to novel poses, the Gaussians tend to produce several undesirable artifacts such as blurred texture, incorrect colors or floating ellipsoid; see Fig 1.

To address the limitations of existing Gaussian-based avatar methods on clothed upper-body, we argue that the chest and shoulders are expected to have relatively simpler geometry and more intricate deformation compared to the head. Therefore, modeling them with regular and 3DMM-driven Gaussians would be unsuitable and is an over-complication of the problem. Instead, a more appropriate and standard approach would be representing their appearance with a high-frequency texture.

In a traditional texture-based rendering pipeline, the texture is first mapped to mesh geometry in the 3D world space via UV mapping, and then rasterized to the 2D image plane in the view space to obtain the pixel color. However, this approach requires a well-defined UV mapping and accurate mesh geometry, which is challenging to obtain from monocular videos alone due to the lack of multi-view correspondences. Besides, compared to faces that share more common characteristics and stronger priors, the appearance of upper bodies can vary dramatically depending on the texture and tightness of the clothes and they hence contain fewer detectable landmarks. Consequently, body 3DMMs such as SMPL (Loper et al., 2015) fail to provide geometry accurate enough for this purpose.

Hence, we propose to bypass the mapping from texture space to world space, and instead use a sparse set of Gaussians as "anchors" to define a direct neural warping field from a canonical 2D texture space, which consists of a coarse RGB texture and a fine neural texture, to the image plane. As the tracking of body 3DMM tends to be inaccurate due to the lack of landmarks, we only transform anchor Gaussians together with the head Gaussians via a head FLAME 3DMM (Li et al., 2017) through Linear Blend Skinning (LBS). The transformed anchor Gaussians are used as soft constraints of the texture warping represented by a coordinate-based MLP, which is optimized together with the neural texture, regular Gaussians, and the anchor Gaussians. As the resolution of the neural texture is not limited by the number of Gaussians or the density control scheme, we can easily learn the high-frequency textures with sharp details on the clothes and avoid the common artifacts exhibited in Gaussian rendering under novel poses; see Fig 1.

To maintain a competitive rendering speed with Gaussian Splatting and enable real-time interactive applications, we additionally propose a method to remove the neural warping field and neural texture in the model and allow inference of reconstructed avatars at novel poses without any MLP queries. This accelerated inference effectively increases the rendering speed from 70 FPS to around 130 FPS, which surpasses the rendering speed of plain Gaussian Splatting avatars for subjects with high-frequency clothes.

We evaluate the proposed method with various casual monocular videos collected using smartphones or from the Internet. Compared to state-of-the-art methods which incorporate different representations including neural radiance field, Gaussian Splatting, and point clouds, we show that our approach achieves better performance and robustness for both self-reenactment and cross-reenactment tasks. In summary, our contributions are:

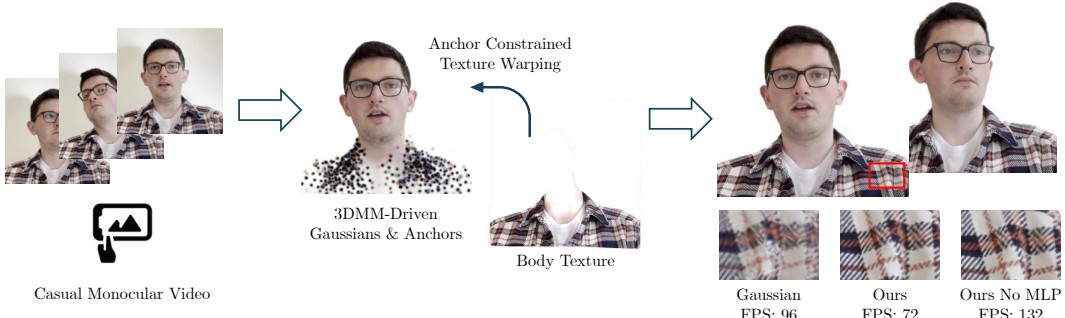

Figure 1: **Gaussian Head & Shoulders** reconstructs 3DMM-driven upper body avatars from casual monocular videos. By utilizing a high-frequency body neural texture which is warped using a neural texture warping field constrained by a set of sparse anchor Gaussians, we can learn sharp details of the cloth texture with highly efficient rendering speed.

- We propose a novel approach that maps intricate texture to the image plane via a sparse set of anchor Gaussians driven by LBS with 3DMM. This allows accurate and robust modeling of high-fidelity clothed chest and shoulders with less number of Gaussians.

- We propose a method to remove the MLP in our method at inference time to prevent any costly queries when rendering with novel poses and expressions and reach a rendering speed of around 130 FPS.

## 2 RELATED WORKS

**Neural Head Avatars** The recent advancement in neural 3D implicit and explicit representations has sparked a surge of methodologies within the field of controllable 3D head avatars. Among these approaches, a prominent family of methods involves the reconstruction of a 5D neural radiance field, manifested through various forms such as pure MLP (Gafni et al., 2021; Wang et al., 2021; Kirschstein et al., 2023), hash grid latents (Xu et al., 2023; Gao et al., 2022; Zielonka et al., 2022; Xu et al., 2023; Dhamo et al., 2023; Xiang et al., 2024; Saito et al., 2024; Chen et al., 2023), and 3D Gaussians (Wang et al., 2024; Zhao et al., 2024). Another set of methods utilizes more explicit representations such as deformable meshes with neural textures (Grassal et al., 2021; Zheng et al., 2022; Buehler et al., 2021; Gropp et al., 2020; Khakhulin et al., 2022; Kim et al., 2018) and point clouds (Zheng et al., 2023). With the most recent Gaussian Splatting techniques, the head avatars reconstructed from monocular videos have already reached high fidelities. However, many methods simplify the problem by reconstructing only the head and neck part, resulting in a head-only reconstruction that is not suitable for many applications. Several methods have attempted to also model the chest and shoulders to provide a more immersive user experience (Zheng et al., 2022; Zhao et al., 2024; Wang et al., 2024; Zheng et al., 2023), but they are limited to simple clothes with plain colors, and cannot handle the movements in the upper body in the video.

**Neural Full-Body Avatars** Several works have tried to reconstruct a controllable full-body neural avatar from multi-view or monocular videos (Liu et al., 2024; Shao et al., 2024; Svitov et al., 2024; Li et al., 2024; Lei et al., 2023; Kocabas et al., 2023; Hu et al., 2024b; Jiang et al., 2022). Due to the highly articulated nature of human bodies, they tightly rely on body 3DMMs to deform the neural body representation via LBS and hence fail to faithfully capture subjects with complicated or loose clothing as those cannot be modeled with existing body 3DMMs. Besides, they typically focus on the overall quality of the torso and limbs, and hence tend to present non-trivial artifacts when reconstructing and re-animating an avatar that has a tight focus around the head and shoulder regions.

## 3 METHOD

Given a monocular video featuring a talking subject with various expressions and head poses, our goal is to reconstruct a high-fidelity and animatable avatar including the head and clothed upper body. As illustrated in Fig 2, our method jointly optimizes 1) a set of standard 3D Gaussians (Kerbl et al., 2023) which tightly follow the transformation of 3DMM via LBS to represent the head region,

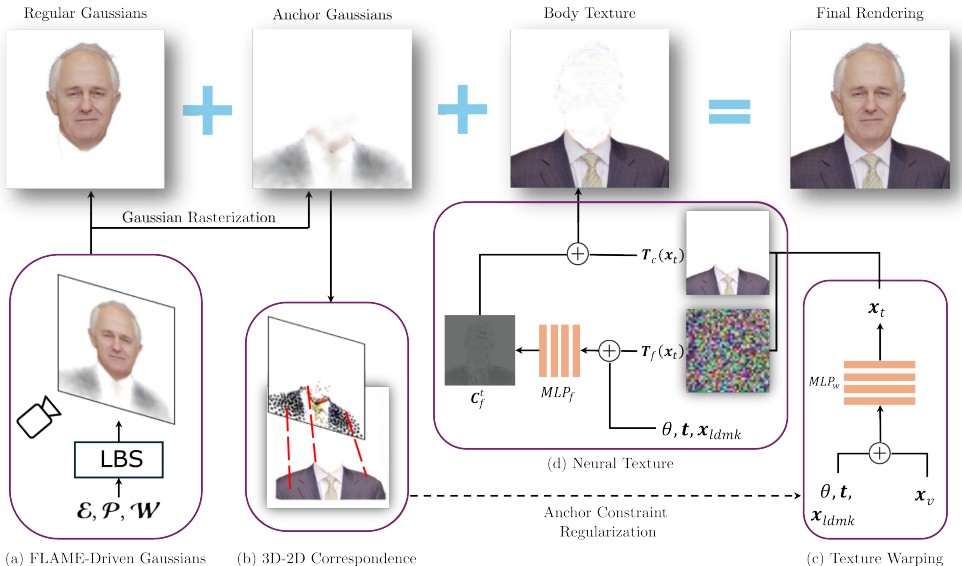

Figure 2: **Method.** (a) We utilize a set of standard head Gaussians and anchor Gaussians driven by LBS with the FLAME model. (b) Anchor Gaussians are initialized with a set of corresponding target coordinates in the texture space. This 3D-2D correspondence is used to constrain (c) a neural texture warping field that maps each pixel on the image plane $\mathbf{x}_v$ to a pixel in the texture space $\mathbf{x}_t$. (d) We then sample in the texture space to fetch the coarse texture $\mathbf{T}_c$ and latent texture $\mathbf{T}_f$, which is parsed by an MLP to obtain pose-dependent fine texture $\mathbf{C}_f^t$. Both coarse and fine textures are then combined to form a body texture, which is blended with other Gaussians through alpha compositing to form the final rendering.

2) a set of sparse anchor Gaussians spawning over the clothed body, and 3) a learnable neural texture with pose-dependent neural texture warping field constrained by the anchor Gaussians to represent the clothed body with sharp details and high robustness.

## 3.1 PRELIMINARY- GAUSSIAN SPLATTING

3D Gaussian Splatting is a volumetric representation that utilizes a dense set of anisotropic Gaussians with varying opacity and view-dependent radiance to represent 3D geometry and appearance. Each Gaussian is described with four parameters: position (Gaussian mean) $\boldsymbol{\mu}$, 3D covariance matrix $\boldsymbol{\Sigma}$, opacity $\alpha$ and Spherical Harmonic (SH) coefficients $\mathbf{SH}$ for computing view-dependent RGB color. For ease of optimization, the covariance matrix is further decomposed into a scaling matrix $\mathbf{S}$, stored as a scaling vector $\mathbf{s}$, and a rotation matrix $\mathbf{R}$, stored as a quaternion vector $\mathbf{q}$. The covariance matrix is obtained as: $\boldsymbol{\Sigma} = \mathbf{R}\mathbf{S}\mathbf{S}^T\mathbf{R}^T$.

To render 3D Gaussians to RGB images, their means are projected onto 2D image plane with standard projective transformation, while the projected covariance matrix is obtained by $\boldsymbol{\Sigma}' = \boldsymbol{J}\boldsymbol{W}\boldsymbol{\Sigma}\boldsymbol{W}^T\boldsymbol{J}^T$, where $\mathbf{W}$ is the world to camera transformation and $\mathbf{J}$ is the Jacobian approximating the projective transformation (Zwicker et al., 2001). The rendered RGB color at each pixel is then obtained through:

$$\mathbf{C}(\mathbf{x}) = \sum_{i \in N} \mathbf{c}_i \alpha_i^*(\mathbf{x}) \prod_{j=1}^{i-1} (1 - \alpha_j^*(\mathbf{x})), \tag{1}$$

$$\alpha_i^*(\mathbf{x}) = \alpha_i \exp\left(-\frac{1}{2}(\mathbf{x} - \boldsymbol{\mu}_i')^T \boldsymbol{\Sigma}'^{-1}(\mathbf{x} - \boldsymbol{\mu}_i')\right), \tag{2}$$

where $\mathbf{x}$ is the 2D pixel coordinate, $\mathbf{c}_i$ is the view-dependent RGB radiance of i-th Gaussian on the ray obtained from SH function, $\alpha_i$ and $\boldsymbol{\mu}_i'$ are the opacity and projected 2D mean of the i-th Gaussian respectively.

## 3.2 FLAME-DRIVEN HEAD GAUSSIANS

As the face region contains highly distinguishable characteristics and can be described accurately with parametric head 3DMM such as FLAME (Li et al., 2017), we directly utilize standard 3D Gaussians that are deformed with parametric 3DMM via neural LBS to represent the head part (Zheng et al., 2023; Zhao et al., 2024). Specifically, we learn personalized FLAME expression and pose blendshapes and LBS weights through a small 3D coordinate-based MLP for each Gaussian: $\mathcal{E}, \mathcal{P}, \mathcal{W} = \text{MLP}_d(\boldsymbol{\mu})$, where $\mathcal{E} \in \mathbb{R}^{n_e \times 3}$ are the expression blendshapes, $\mathcal{P} \in \mathbb{R}^{n_p \times 9 \times 3}$ are the pose blendshapes, $\mathcal{W} \in \mathbb{R}^{n_j}$ are the LBS weights corresponding to each of the $n_j$ bones. Following (Hu & Liu, 2023), we use the standard skinning function LBS to obtain the rotation $\boldsymbol{R}$ and translation $\boldsymbol{T}$ for each Gaussian, and apply them to get the Gaussian mean $\boldsymbol{\mu}^d$ and covariance $\boldsymbol{\Sigma}^d$ in the 3D view space:

$$\boldsymbol{R}, \boldsymbol{T} = \text{LBS}(\boldsymbol{B}_{\mathcal{P}}(\theta; \mathcal{P}) + \boldsymbol{B}_{\mathcal{E}}(\psi; \mathcal{E}), \mathbf{J}(\psi), \theta, \mathcal{W}), \tag{3}$$

$$\boldsymbol{\mu}^d = \boldsymbol{R}\boldsymbol{\mu} + \boldsymbol{T}, \ \boldsymbol{\Sigma}^d = \boldsymbol{R}\boldsymbol{\Sigma}\boldsymbol{R}^T, \tag{4}$$

where $\mathbf{J}$ is the joint regressor in FLAME, and $\boldsymbol{B}_{\mathcal{P}}$ and $\boldsymbol{B}_{\mathcal{E}}$ are linear combination of blendshapes based on per-frame coefficients $\theta$ and $\psi$ that control the head animation. They can then be rendered with a standard Gaussian rasterization pipeline in Eq 1.

## 3.3 3D-2D CORRESPONDENCE VIA ANCHOR GAUSSIANS

3D Gaussian Splatting has shown promising performance and robustness in reconstructing 3D geometry and appearance from RGB images. However, they suffer from a significant constraint – each individual Gaussian can only represent a spatially invariant color under a fixed viewing direction, hence a vast number of Gaussians is required to represent objects with detailed textures, regardless of the actual complexity of the geometry. A naive application of Gaussian Splatting therefore fails to capture the fine details of the upper body with complex textures and intricate deformation, and results in blurry details and floating artifacts under challenging poses.

We hence propose to learn a high-frequency texture in canonical texture space, and use a sparse set of Gaussians as anchors to guide the warping between texture space and image plane. As such, we only need a small number of Gaussians and a texture with per-pose warping to represent a clothed body with arbitrarily complicated textures. Since anchor Gaussians themselves do not need to exactly represent the high-frequency appearance, we can model them as a simplified version of regular Gaussians: they only use view-independent RGB colors, are isotropic Gaussians with quaternion fixed at $(1, 0, 0, 0)$, and are excluded from the density control and therefore are not split, cloned, or pruned. To prevent them from becoming trivial in rendering, their opacity and size are clamped to be no smaller than 0.05 and 0.0001 respectively.

The anchor Gaussians are initialized as follows: after a short warm-up period that only trains plain Gaussian, we first reproject all Gaussian means onto the image plane of a canonical training frame, and filter out Gaussians that are located around the head region based on semantic masks. We then use farthest point sampling (Qi et al., 2017) to select $N_a = 1024$ Gaussians as anchor Gaussians. The first SH basis is converted to RGB values and the anchor scales in three directions are averaged to form a single scale for the anchor Gaussians. We then obtain a sparse set of anchor Gaussians, as well as their projected 2D means $\hat{\mathbf{x}}_i^v$ on the image plane (2D view space) of the canonical frame:

$$\hat{\mathbf{x}}_i^v = \mathbf{P}(\hat{\boldsymbol{\mu}}_i^d), \tag{5}$$

where $\mathbf{P}$ is the camera projective transformation, $\hat{\boldsymbol{\mu}}_i^d$ is the 3D Gaussian mean of the $i$-th anchor Gaussian transformed to 3D view space with LBS. To build the correspondence between anchor Gaussians and texture space coordinates, we assume that the mapping between the 2D image plane of the canonical frame and the texture space is an identity mapping. We can hence define a function $f_{anchor}(i)$ as a fixed correspondence between the $i$-th 3D anchor Gaussian mean and its target 2D pixel coordinate in texture space:

$$f_{anchor}(i) := \mathbf{I}(\hat{\mathbf{x}}_i^v), \tag{6}$$

where $\mathbf{I}$ is the identity function to map 2D image plane coordinates to texture space. Note that $f_{anchor}(i)$ is fixed after initialization and does not update with further optimization of $\hat{\boldsymbol{\mu}}_i$. Such correspondences will later be used to constrain the pose-dependent texture warping, as will be detailed in Sec 3.6.

### 3.4 Neural Texture and Texture Warping

We use a trainable neural texture in canonical space with a pose-dependent neural texture warping field to represent the part of the avatar with relatively simple overall geometry and complicated appearances, i.e., the clothed shoulder and chest. In a traditional textured mesh rendering pipeline, the texture is first mapped to the mesh triangles through a pre-defined UV mapping, and the meshes are then rasterized to find the first intersections with the camera rays. Those first intersections therefore establish a mapping between texture space and image plane. However, this approach is not applicable without accurate surfaces and well-defined UV mapping. We instead propose to bypass the intermediate step and learn a per-pose warping that directly maps pixel coordinates on image plane $\mathbf{x}_v$ to the texture coordinates $\mathbf{x}_t$ for texture fetching. Specifically, the warping field is represented using a coordinate-based MLP:

$$\Delta_{\mathbf{x}} = \mathrm{MLP}_w \left( \gamma(\mathbf{x}_v), \gamma(\theta), \gamma(\mathbf{t}), \gamma(\mathbf{x}_{ldmk}) \right),\tag{7}$$

where $\gamma$ is the positional encoding (Mildenhall et al., 2020), $\theta$ is the FLAME pose parameters containing head and neck rotations, $\mathbf{t}$ is the camera position, $\mathbf{x}_{ldmk}$ is 2D body landmarks for neck, left and right shoulders. The corresponding texture coordinate is obtained as $\mathbf{x}_t = \mathbf{x}_v + \Delta_{\mathbf{x}}$.

Our optimizable texture includes a coarse texture $\mathbf{T}_c$ with 3 channels and a latent texture $\mathbf{T}_f$ with $D_t$ channels. Both textures have sizes of $[H + 2P, W + 2P]$, where $H, W$ are the image height and width, $P$ is the padding size which we empirically set to 50 to account for body parts that move in and out in the video sequence. The latent texture $\mathbf{T}_f$ is passed to an MLP to obtain pose-dependent appearances such as brightness changes on the clothes:

$$\mathbf{C}_f^t(\mathbf{x}_t) = \mathrm{MLP}_f \left( \mathbf{T}_f(\mathbf{x}_t), \gamma(\theta), \gamma(\mathbf{t}), \gamma(\mathbf{x}_{ldmk}) \right),\tag{8}$$

where $\mathbf{T}_c(\mathbf{x}_t), \mathbf{T}_f(\mathbf{x}_t)$ are coarse and latent texture sampled at 2D coordinate $\mathbf{x}_t$ via bilinear interpolation. The textured pixel color at the coordinate $\mathbf{x}_v$ is therefore obtained as $\mathbf{C}^t(\mathbf{x}_v) = \mathbf{T}_c(\mathbf{x}_t) + \mathbf{C}_f^t(\mathbf{x}_t)$.

By constraining with the correspondences between deformable anchor Gaussians and their fixed projections on 2D texture space, we can learn accurate and effective texture warping for various body movements including translation, rotation, and depth-based (in-and-out) motions; see Fig 3.

### 3.5 Rendering

To this end, we have a hybrid representation that includes 3D regular Gaussians that represent the head of the avatar, 3D anchor Gaussians that sparsely span over the body region, and a 2D neural texture for the body. To render all of them together for joint optimization, we simply use alpha blending:

$$\mathbf{C}^*(\mathbf{x}_v) = \underbrace{\hat{\mathbf{C}}(\mathbf{x}_v)}_{\text{Anchor Gaussians}} + \underbrace{(1 - \hat{\alpha}(\mathbf{x}_v))\mathbf{C}(\mathbf{x}_v)}_{\text{Head Gaussians}} + \underbrace{(1 - \hat{\alpha}(\mathbf{x}_v))(1 - \alpha(\mathbf{x}_v))\mathbf{C}^t(\mathbf{x}_v)}_{\text{Body Texture}},\tag{9}$$

where $\hat{\mathbf{C}}(\mathbf{x}_v), \mathbf{C}(\mathbf{x}_v)$ are the rendered RGB color of anchor Gaussian and regular Gaussian, $\hat{\alpha}(\mathbf{x}_v), \alpha(\mathbf{x}_v)$ are the total alpha of anchor Gaussian and regular Gaussian at pixel $\mathbf{x}_v$ respectively.

Note that our rendering process always renders anchor Gaussians in front of the regular Gaussians regardless of their actual positions. Though not physically realistic, we designed this rendering order so anchor Gaussians are always non-trivial and never occluded by regular Gaussians.

### 3.6 Optimization

The optimization is split into three different stages: anchor warm-up stage, main optimization stage, and texture refinement stage. In the anchor warm-up stage, neither anchor Gaussians nor body texture is applied, only the regular Gaussians are rendered and optimized. The purpose of this stage is to move Gaussians to roughly spawn over the area of interest including both head and body. At the end of this stage, we initialize anchor Gaussians from regular Gaussians using the method described in Sec 3.3. In the second stage, we render all of the regular Gaussians, anchor Gaussians, and the textured body with alpha compositing described in Eq 9 and jointly optimize them together. In the last stage, to recover faithful appearance for the body texture and enhance its robustness under novel

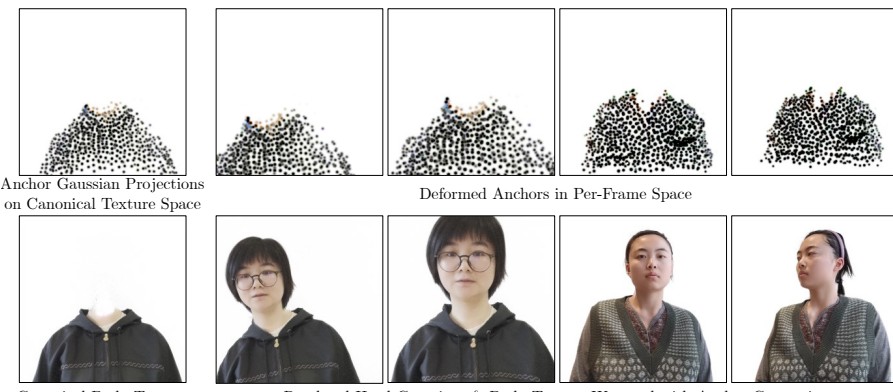

Anchor Gaussian Projections on Canonical Texture Space

Deformed Anchors in Per-Frame Space

Canonical Body Texture

Rendered Head Gaussians & Body Texture Warpped with Anchor Constraint

Figure 3: **Anchor Warping.** The anchors are initialized with corresponding projections on canonical texture space. When anchors are deformed via LBS to model the per-frame body movement, they map to the same projections in texture space and hence establish correspondences for body texture warping.

poses, we remove anchor Gaussians from the rendering pipeline, i.e., we set $\hat{\mathbf{C}}$ and $\hat{\alpha}$ to $0$ in Eq 9., and freeze everything else except for the neural texture, texture warping field, and opacity and SH of regular Gaussians.

Following (Zheng et al., 2023; 2022), the training losses include standard MSE RGB loss $\mathcal{L}_C = MSE(\mathbf{C}^* - \mathbf{C}^{GT})$, and a FLAME regularization that encourages the FLAME blendshapes and LBS weights predicted for each Gaussian stay close to the pseudo ground truth $\tilde{\mathcal{E}}_i, \tilde{\mathcal{P}}_i, \tilde{\mathcal{W}}_i$ obtained from the nearest FLAME vertex:

$$\mathcal{L}_{flame} = \frac{1}{N} \sum_{i=1}^{N+N_a} (\lambda_{\mathcal{E}} |\mathcal{E}_i - \tilde{\mathcal{E}}_i|_2 + \lambda_{\mathcal{P}} |\mathcal{P}_i - \tilde{\mathcal{P}}_i|_2 + \lambda_{\mathcal{W}} |\mathcal{W}_i - \tilde{\mathcal{W}}_i|_2). \tag{10}$$

During main optimization stage, we additionally include a VGG feature loss (Johnson et al., 2016; Simonyan & Zisserman, 2015) $\mathcal{L}_{VGG} = |\mathbf{F}_{vgg}(\mathbf{C}) - \mathbf{F}_{vgg}(\mathbf{C}^{GT})|$, and a head mask regularization to encourage regular Gaussians to stay only within the head region and allow the body texture to be trained properly without being occluded:

$$\mathcal{L}_{head} = MSE(max(\alpha - \alpha_{head}, 0)), \tag{11}$$

where $\alpha_{head}$ is the alpha mask of the head region obtained with matting pre-processing and semantic mask. We also include an L1 regularization on the 2D neural warping field to encourage a clean background to be learned in the neural texture, as well as an L1 loss to slowly decrease the opacity of anchor Gaussians to allow the body texture to be trained properly:

$$\mathcal{L}_{warp} = \frac{1}{HW} \sum_{i=1}^{HW} |\Delta_{\mathbf{x}_i}|, \quad \mathcal{L}_{\hat{\alpha}} = \frac{1}{N_a} \sum_{i=1}^{N_a} |\hat{\alpha}_i|. \tag{12}$$

Finally, we include an anchor loss as a soft constraint of the per-pose texture warping:

$$\mathcal{L}_{anchor} = \frac{1}{N_a} \sum_{i=1}^{N_a} (f_{anchor}(i) - (\hat{\boldsymbol{x}}_i^v + \Delta_{\hat{\boldsymbol{x}}_i^v}))^2, \tag{13}$$

i.e., for each anchor Gaussian, we first transform it to 3D view space via LBS, and then project it onto the image plane to obtain its 2D mean $\hat{\boldsymbol{x}}_i^v$ via Eq 5. $\hat{\boldsymbol{x}}_i^v$ is then warped by the neural warping field $\text{MLP}_w$ to obtain the corresponding coordinate in the texture space, which is optimized to match the anchor correspondence defined during initialization.

In the third stage, we remove the regularization losses including $\mathcal{L}_{head}, \mathcal{L}_{warp}$ and $\mathcal{L}_{\hat{\alpha}}$.

The total training objectives for each of the three stages are as follows:

$$\mathcal{L}_1 = \mathcal{L}_C + \mathcal{L}_{flame}, \tag{14}$$
$$\mathcal{L}_2 = \mathcal{L}_1 + \lambda_{VGG}\mathcal{L}_{VGG} + \lambda_{head}\mathcal{L}_{head} + \lambda_{warp}\mathcal{L}_{warp} + \lambda_{\hat{\alpha}}\mathcal{L}_{\hat{\alpha}} + \lambda_{anchor}\mathcal{L}_{anchor},$$
$$\mathcal{L}_3 = \mathcal{L}_1 + \lambda_{VGG}\mathcal{L}_{VGG} + \lambda_{anchor}\mathcal{L}_{anchor}. \tag{15}$$

### 3.7 Accelerated Rendering with No MLP Queries

One of the main advantages of Gaussian Splatting is its highly efficient rendering speed, which enables many real-time and interactive applications. To take full use of this advantage, we propose an accelerated version of our method that requires no MLP queries at inference time. Specifically, after training the model, we first cache the output of $\mathrm{MLP}_d$ for all head Gaussians and anchor Gaussians, then cache the view-dependent fine texture by querying the fine texture MLP $\mathrm{MLP}_f$ conditioned on the same canonical training frame which was previously used to initialize the anchor Gaussians. The queried fine texture colors are added to the coarse color to make a non-neural RGB texture. To deal with potential noise created by the fine texture MLP at the corners of the texture, we use an off-the-shelf background segmentation network (Chen et al., 2017) to compute a coarse mask and clean all the pixels outside of the mask; we show the necessity of this step in the supplementary. To replace the neural warping field $\mathrm{MLP}_w$ that warps image plane coordinates to texture space, we rely on the correspondence between anchor Gaussians and texture space coordinates to estimate a homography at inference time. Specifically, we first project all anchor Gaussians to the image plane of the canonical training frame, and then remove any anchor Gausians that go beyond the view frustum. To deal with any potential discrepancy between the neural warping field and the anchor correspondences, we update those correspondences based on the prediction of the neural warping field on the current frame:

$$f_{anchor}(i) := \hat{\mathbf{x}}_v^i + \Delta_{\hat{\mathbf{x}}_v^i}. \tag{16}$$

After that, we randomly select 100 training frames and use RANSAC (Fischler & Bolles, 1981) to estimate a homography between the image plane coordinates of anchor Gaussians and their corresponding texture space coordinates, and remove anchor Gaussians that are considered outliers by RANSAC. This effectively removes any anchor deformation that cannot be described by the rigid transformation. Finally, at inference time, we perform LBS on regular head Gaussians and anchor Gaussians. Based on the image plane coordinates of the anchor Gaussians $\hat{\mathbf{x}}_v^i$ and their correspondences $f_{anchor}$, we compute a homography with the least square error via singular value decomposition. The estimated transformation is applied to all pixels on the image plane to find the corresponding non-neural texture, which is then blended with the head Gaussians to form the final rendering. This accelerated inference approach effectively increases the rendering speed from around 70 FPS to 130 FPS.

## 4 Evaluation

**Datasets**   We evaluate different methods on 1 mobile phone sequence from PointAvatar (Zheng et al., 2023), 2 internet sequences from Head2Head dataset (Koujan et al., 2020), and 4 sequences captured with mobile phones. All sequences are preprocessed with DECA (Feng et al., 2021) and a slightly modified landmark fitting process from IMAvatar (Zheng et al., 2022). Additionally, we use DWPose (Yang et al., 2023) to predict 2D landmarks for nose, neck and shoulders, which are then smoothed with One Euro Filter (Casiez et al., 2012).

**Baselines**   We compare our method with four neural head avatar methods based on various representations, including (1) INSTA (Zielonka et al., 2022), which employs a latent hash grid (Müller et al., 2022) combined with NeRF (Mildenhall et al., 2020), (2) PointAvatar (Zheng et al., 2023), which is based on isotropic point clouds, (3) SplattingAvatar (Shao et al., 2024) and (4) GaussianAvatars Hu et al. (2024a), which utilize Gaussian Splatting attached to local space of 3DMM meshes, and (5) GS*, a baseline we implemented by changing the point cloud representation in PointAvatar to Gaussian Splatting, which is similarly deformed via neural LBS.

**Self-Reenactment**   We show the quantitative and qualitative results of the self-reenactment task in Tab 1 and Fig 4. Our full version demonstrates superior reconstruction performance compared to existing baselines, especially for subjects with intricate cloth textures. Our No MLP version does not consistently achieve better PSNR when compared to existing baselines, as it is unable to render pose-dependent appearance changes and intricate cloth deformation. However, we note that it consistently achieves better LPIPS, demonstrating that our No MLP version can still generate realistic and faithful renderings. This discrepancy among different metrics arises because of the high sensitivity of PSNR to small misalignments in the cloth texture (Park et al., 2021). As a result, PSNR

|  | full | | | head | | |
|---|---|---|---|---|---|---|
|  | PSNR↑ | SSIM↑ | LPIPS↓ | PSNR↑ | SSIM↑ | LPIPS↓ |
| INSTA | 20.59 | .781 | .236 | 29.80 | .936 | .059 |
| SplattingAvatar | 21.00 | .774 | .274 | 28.29 | .925 | .069 |
| PointAvatar | 25.21 | .851 | .102 | 30.64 | .943 | .042 |
| FlashAvatar | 21.88 | .808 | .127 | 30.33 | .946 | .039 |
| GaussianAvatars | 20.97 | .801 | .193 | 29.285 | .941 | .046 |
| GS* | 25.94 | .854 | .095 | 31.81 | .947 | .036 |
| Ours | 26.80 | .875 | .070 | 33.06 | .959 | .028 |
| Ours (No MLP) | 25.47 | .859 | .074 | 32.54 | .956 | .029 |

Table 1: **Quatitative evaluation of self-reenactment task** We color the best and second-best methods. Our full method achieves much better performance compared to existing baselines. While Ours (No MLP) achieves slightly lower PSNR, which is known to be over-sensitive to small misalignments and prefers blurry results (Park et al., 2021), we show it achieves better LPIPS than existing methods.

|  | FPS | #GS | FPS | #GS |
|---|---|---|---|---|
|  | 003 | | 004 | |
| FlashAvatar | 134 | 13453 | 137 | 13453 |
| GS* | 141 | 163830 | 159 | 125521 |
| Ours | 70 | 58701 | 71 | 39549 |
| Ours (No MLP) | 129 | 58701 | 134 | 39549 |
|  | 005 | | 007 | |
| FlashAvatar | 125 | 13453 | 126 | 13453 |
| GS* | 96 | 317968 | 131 | 191431 |
| Ours | 72 | 50708 | 69 | 52910 |
| Ours (No MLP) | 132 | 50708 | 127 | 52910 |

Table 2: **Performance measure.** We report rendering FPS and the number of Gausssians for each method. The rendering speed of our no MLP version surpasses pure Gaussian implementation for subjects wearing extremely high-frequency cloth.

tends to prefer blurry reconstruction over sharp but slightly misaligned results. Notably, although we did not include specific treatments for the head region, better modeling of the body also leads to better face reconstruction. The qualitative evaluation in Fig 4 demonstrates that both versions of our method can learn sharper and more robust body texture compared to existing methods.

**Cross-Reenactment** For the cross-identity reenactment task, we render the reconstruction of the original identity with FLAME expressions and poses from the source subject. With the full version of our method, we apply an additional Euclidean transformation after warping the image plane coordinates with the MLP. This is to ensure the body texture is always aligned with the head Gaussians under novel poses; see Fig 6. The Euclidean transformation is simply determined by fitting the MLP warped image plane coordinates of the anchor Gaussians and their target coordinates in the texture space. To deal with potential artifacts caused by coordinates warped to unseen corner parts in the texture, we apply the same appearance distillation process and remove the fine texture MLP. The No MLP version is applied the same way as in the self-reenactment task.

In addition to the improvement over the body texture, we observe that avatars reconstructed with our approach often give more accurate and faithful expression control, as shown in Fig 5. We deduce that this is because the 3DMM-driven Gaussians only need to model the head region, leading to a more accurate reconstruction of the head model and more reliable LBS weights and expression and pose blendshapes predicted by the LBS network.

**Ablation** We show the effectiveness of the anchor constraint $\mathcal{L}_{anchor}$, test-time Euclidean transformation and warp loss $\mathcal{L}_{warp}$ in Fig 6. Even for subjects with only slight movement in the upper body, anchor constraint is still needed to learn sharp and accurate cloth texture. Besides, without anchor Gaussians and test time Euclidean transformation, the body texture is unable to align with the head Gaussians under novel poses. The warp loss $\mathcal{L}_{warp}$ is needed to prevent the neural warping field from mapping the background pixel to an arbitrary white pixel in the texture space. As anchor Gaussians only exist within the body region, the additional Euclidean transformation computed from anchor correspondences would significantly distort the background pixels, causing severe artifacts as shown in Fig 6 (b). Additional ablation results can be found in the supplementary.

**Rendering Efficiency** We report the number of Gaussians and the rendering speed for pure Gaussian implementation GS*, Ours, and Ours (No MLP) in Tab 2. The rendering speeds are tested on an RTX4080 Ti. For subjects wearing complicated clothes, the number of Gaussians required to model the high-frequency cloth texture significantly increases for pure Gaussian implementation, hence slowing down the rendering speed, whereas our method only models the head region with Gaussians and hence requires a much fewer number of Gaussians. The rendering speed of our no

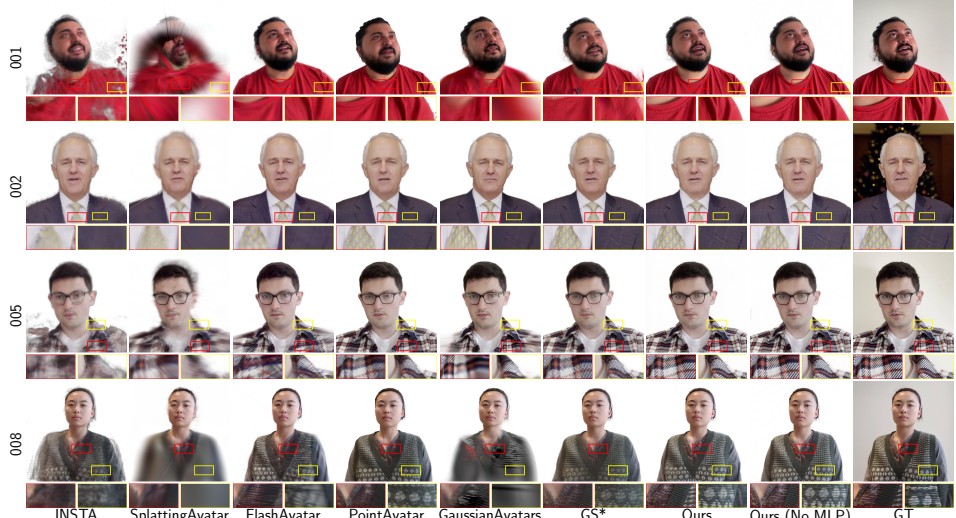

Figure 4: **Qualitative comparison of self-reenactment task.** We show that both of our full version and No MLP version can recover a more accurate and robust body texture, even under extreme poses and high-frequency cloth textures. More results in the Supplementary 7.

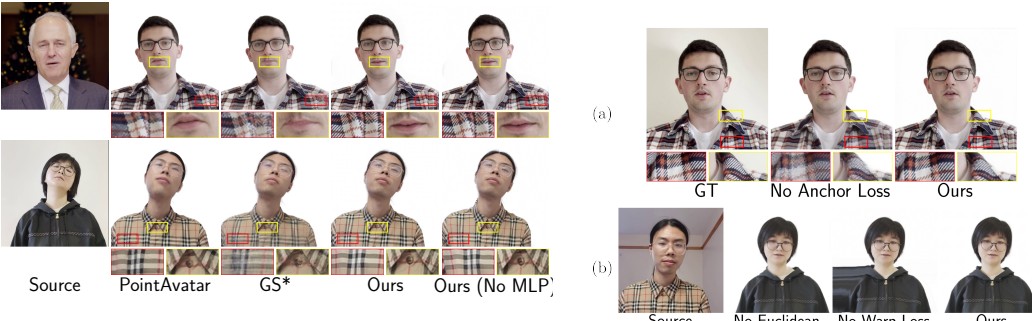

Figure 5: **Qualitative evaluation of cross-identity reenactment.** Our method leads to both better cloth texture and more accurate expression, as LBS network only focuses on the head region in our approach. More results in the Supplementary 8.

Figure 6: **Qualitative ablation for self-reenactment (a) and cross-reenactment (b).**

MLP version even surpasses pure Gaussian implementation for subject 005, who wears cloth with a very high-frequency texture.

## 5 CONCLUSION

We present Gaussian Head & Shoulders, a method that reconstructs high-quality and animatable upper body avatars including head, chest and shoulders. By utilizing high-frequency neural texture to represent the clothed body, we are able to model sharp and robust cloth details and significantly reduce the number of Gaussians needed to represent a subject. By constraining the texture warping with a sparse set of anchor Gaussians, the body texture is accurately mapped to the correct position even under unseen poses. By caching the neural texture and replacing the neural warping field with a projective transformation estimated using anchor correspondences, we significantly improve rendering speed and reach over 130 FPS at novel poses, surpassing the rendering speed of pure Gaussian implementation for subjects with complicated cloth textures.

**Limitation.** Although our method can learn faithful texture for the shoulder and chest, it cannot handle arm and hand motions, which would require specific prior and representation such as SM-PLX (Loper et al., 2015).

ACKNOWLEDGEMENTS

This work was supported by a UKRI Future Leaders Fellowship [grant number G104084].

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
