# OpenReview forum: "Gaussian Head & Shoulders: High Fidelity Neural Upper Body Avatars with Anchor Gaussian Guided Texture Warping"
_ICLR.cc/2025/Conference — ICLR 2025 Poster_

### Official Review · Reviewer_Q3T9 · 2024-10-16

**Soundness:** 2
**Presentation:** 3
**Contribution:** 2
**Rating:** 8
**Confidence:** 4

**Summary:**

This work present the way of rendering shoulders in addition to the human faces. In general, the author propose the following methods:
(1) to use anchor gaussians to represent the shoulder region only.
(2) use neural texture to refine the shoulder represent and make it robust to body motions including translation, rotation, and depth-based motions;
(3) Leverage 3D-to-2D correspondence mapping as the regularization of the shoulder learning.
(4) an option without using MLP for fast rendering

The author conducts many experiments to validate their method.

**Strengths:**

The overall writing is easy to following and well-written.

It seems like compared with conventional method, which ignore the shoulder but only focus on face, this model presents better performance and more robust to shoulder renderings (like compare with PointAvatar and GS*)

**Weaknesses:**

Concerns of Comparisons:

1. Some works already tries to explore the joint face and shoulder rendering including:
   a. Real3D-Portrait, which uses FOMM for the shoulder motion reenactment and finally merge the face motion and shoulder motion together.
  b. StyleAvatar, a 2D human face rendering technique which includes the face, shoulder and background together

The author did not include these two into the comparison. Is it possible to compare with this pipepile with FOMM/TPS-based methods used in Read3D-Portrait / Torso NeRFs used previously in GeneFace to see whether this method is more effective? In addition, using neural texture as an whole image enhancement method is also covered in StyleAvatar, but slightly different: this work tries to enhance the shoulder only while StyleAvatar's neural texture is for the whole image. It will be good if the authors can present very strong argument regarding this given my evaluation might be partially wrong.

Data:
The videos in the supplementary material do not present very large shoulder motions. I am not quite sure whether the model is robust in rendering the shoulder. However, I do see for slight shoulder motions, the model's performance is good.

Experiment:
I did not see the ablations for the main method part. The proposed methods include introducing the anchor gaussians for shoulder only, the 3d-2d correspondence, the neural texture compared with the baseline GS and PointAvatar. I will defer this part in the questions.

**Questions:**

1. Can the authors provide the novel view synthesis for rendering the shoulder and also present the volumetric representation of the shoulder areas to let readers know how robust the model is for shoulder rendering?

2. What exactly each module brings or benefits for the shoulder rendering? I think the following are possible ablations:
(a) completely remove anchor gaussian and the 3D-2D correspondence, only relying on neural texture like in StyleAvatar
(b) exclude Neural texture, use anchor gaussian only

3. For the anchor gaussian, how the deformation achieved? If it is relying on LBS, due to that LBS is based on flame mesh that only contains the head and neck, there does not exist a real LBS for the shoulders. How should we understand the relationship between the deformation of shoulder gaussian point and neural texture? Is it a refinement of potential noisy deformed gaussian points?

---

> ### Author Response · Authors · 2024-11-19
>
> ## 1. Additional Comparisons
>
> We show additional comparisons with Real3D-Portrait in Fig. 16 and results from StyleAvatar in Fig. 18. Real3D-Portrait produces less faithful and accurate reenactment results, and StyleAvatar quickly degenerates after 10K iterations. We suppose it is because StyleAvatar is designed and evaluated on videos with a much tiger framing, where the shoulders are hardly visible. Therefore, in our setting its performance becomes less stable.
>
> The pipelines of both methods are similar to ours in the sense that they try to handle the head in 3D using 3DMM, whereas the body is handled using 2D images or 2D latent. However, one key difference is that they utilize CNN models without any direct constraint except for the end-to-end reconstruction loss, whereas our approach incorporates Anchor Gaussians to establish a constraint between 3D LBS and 2D neural wrapping. This allows us to model large body translations and rotations as shown in Fig. 3 even without any 3DMM annotation for the body.
>
> ## 2. Large Shoulder Motions
>
> To remain a fair comparison with our baselines which only use FLAME head 3DMM without extra bones for the body or shoulders, we adapted the same setting and used FLAME only for deforming the Anchor Gaussians. Therefore, our method can model overall body movements such as translation and rotations, but do not model individual shoulder movements such as shrugging. Please note that adding support to shoulder movement modeling by adding bones from SMPL should be simple and realistic to achieve. We chose not to do so as our focus is on addressing the limitation of high-frequency texture representation with Gaussians, rather than supporting shoulder movements by changing the 3DMM.
>
> ## 3. Novel View Synthesis of Shoulders
>
> We believe Reviewer Q3T9 misunderstood some of the key aspects of the proposed method. We would like to correct that the shoulder is modeled in 2D and hence has no volumetric representation. There are only approximately 300 Anchor Gaussians sparsely located around the body area to provide anchor loss as a constraint for 2D neural texture warping. We show novel view synthesis results in Sec B.5 and Fig. 15 in the revised version of the paper. Please note that novel view rendering with extrapolated views is challenging for all neural avatar methods from monocular video. Specifically, methods such as StyleAvatar which models the body in 2D cannot render from novel views at all. In comparison, our use of Anchor Gaussians as a 3D-2D constraint allows us to render novel views that are close to training views while ensuring connectivity between head and body.
>
> ## 4. Ablations
>
> Both of the proposed ablations have already been included in the paper. The proposed ablation (a) is included as the “No Warp Loss” ablation in Fig. 6, Table 5, and Fig. 10. This is because Anchor Gaussians are only used to provide the warp loss via anchor constraints and do not participate in the rendering at inference time, hence removing warp loss is same as disabling Anchor Gaussians. (b) will be similar and worse than our baseline GS* compared in all experiments. This is because Anchor Gaussians are simplified versions of regular Gaussians with additional constraints in their covariance and colors. The model trained with Anchor Gaussians and head Gaussians is upper-bounded by the model using regular Gaussians to represent both head and body.
>
> ## 5. Deformation of Anchor Gaussians
>
> As described in Sec A.3, we follow PointAvatar and IMAvatar to add a virtual static bone to the flame mesh. This would allow Anchor Gaussians to model simple body motions such as rotation and translation of the body with the head, but does not support separated shoulder movements such as shrugging – we hope it also helps to address some concerns regarding the large shoulder motions in the previous question. Please note that our method is fully compatible with additional body or shoulder bones from SMPL, we did not use this setting to remain a fair comparison to our baselines, as our focus is on addressing the limitation of high-frequency texture representation with Gaussians.
>
>
> Based on the comments we received, we believe Reviewer Q3T9 has several potential misunderstandings of our method and contribution. We hope our answers can clarify the misunderstandings and address the concerns raised. We would highly appreciate active discussion from Reviewer Q3T9 to further clarify any questions raised.

---

> > ### Comment · Reviewer_Q3T9 · 2024-11-19
> >
> > Thanks for your prompt response, further explanations of the method design, and  extensive experiments included in the rebuttal period.
> >
> > ---
> >
> > ### Reply to Additional Comparisons:
> > Thank you for your time working on further experiments and comparisons.
> >
> > It is obvious StyleAvatar's training is slow and very unstable, compared with this work, which also highlight the **effectiveness and easy-usage of neural texture augmented for gaussian-based 3D head models** proposed by the authors.
> >
> > For Real3DPortrait, I am concerning the comparison might not be fair. I think the authors might have not conducted further fine-tuning but only leverage the pretrained models for zero-shot setting. To be specific, due to that this work mainly focus on the shoulder regions and the shoulder region representation is still 2D based instead of 3D based, I would like to further see the visual comparison of the shoulder regions in Real3DPortrait with this work (maybe with some comparison videos) to address my question of **whether anchor gaussian and neural texture is a more effective strategy than image-warping based method for 2D shoulder motion synthesis**. The training time, efficiency of method comparison can also be included if possible.
> >
> > I acknowledge this may potentially require too much efforts for the authors within the time strict rebuttal period. It is totally fine if the authors do not address this concern.
> >
> > ---
> >
> > ### Reply to Ablations and Deformation of Anchor Gaussians:
> > Thanks for additional explanations that address my previous misunderstandings of this work.
> >
> > ### Reply to Novel View Synthesis of Shoulders:
> > I appreciate the authors' explanations of novel view synthesis. In the newly added Fig.15, it seems like the visual results represent ranges with very slight camera view changes. I am not sure if adding 2D shoulder synthesis based on anchor gaussian might **add additional restrictions or deteriorate the robustness under novel views by the base model**. In addition, adding some camera augmentations during training might help resolve this problem as in [1], for which they can achieve [-40 degree, +40 degress], view ranges even for monocular settings. It will be more helpful for providing demo videos of novel view synthesis results. Due to this work has not open-sourced, there is no need for a further comparison, but I hope **the finding of this work might give some inspirations to the authors, and may address some limitations** as in Fig.15.
> >
> > [1] Tri2-plane: Thinking Head Avatar via Feature Pyramid [ECCV 2024]
> >
> > ---
> > ### Summary
> >
> > Due to the utilization of 2D representation of shoulder for rendering, it brings the **limitation of this work to achieve novel view synthesis of shoulders**. However, I do appreciate the additional experiments and further explanations by the authors during the discussion period.
> >
> > If the authors can provide demo videos and analysis of **whether adding the shoulder regions will deteriorate the model robustness under novel views**, I am willing to raise the score.
> >
> > If the authors can provide further **analysis and comparisons of 2D image-warping based method, TPS, in Real3DPortrait with anchor gaussian+neural texture and show the training efficiency, or better performance by this work**, this will indicate the anchor gaussian with neural texture can be very effective and inspiring further studies. In case of that, I am willing to further raise the score.

---

> > > ### Author Response · Authors · 2024-11-20
> > >
> > > We sincerely appreciate your prompt response and insightful suggestions, as well as your intent to raise the scores. Here are our replies:
> > >
> > > ## 1. Fair Comparison with Real3D-Portrait
> > >
> > > Yes, the results shown in Fig. 16 are obtained with a zero-shot setting without further fine-tuning. We are currently working on finetuning Real3D-Portrait to provide a more valid comparison and will aim to post the results within the rebuttal period.
> > >
> > > ## 2. Novel View Synthesis
> > >
> > > We have updated the supplementary to include the videos of the novel view synthesis. We included results with larger camera rotation as `supplementary/novel_rot.mp4`, as well as large novel camera translation results as `supplementary/novel_trans_x.mp4`, and `supplementary/novel_trans_y.mp4`.
> > >
> > > Rendering shoulders from novel camera rotation is particularly challenging, mainly because in a monocular video, the subjects **do not rotate their body as much as their head** to allow the shoulders to be visible from a wider range of view angles. Similar to all per-subject fitting methods, our approach trains from scratch for each subject, and therefore cannot effectively synthesize anything that can not be seen in the training video.
> > >
> > > We appreciate the reviewer's recommendation of the augmentation technique from Tri2-plane. We believe it is an effective approach for improving robustness, especially against novel camera translations. We would like to note that, our method already achieves robust synthesis of novel camera translation without such augmentation, as shown in  `supplementary/novel_trans_x.mp4`, and `supplementary/novel_trans_y.mp4` in the supplementary. We believe achieving high quality novel camera rotation rendering of shoulders with this technique would still be challenging, simply because the limited view angles of shoulders in our training dataset.
> > >
> > > Regarding **whether adding the shoulder regions will deteriorate the model robustness under novel views**, we would like to note that, for typical real-world applications, it is not necessary to render shoulders with the same novel camera view for the head. This is because applications such as video conferencing typically only require the head to be rotatable, whereas the body should stay mostly forward-facing. The most critical issue is instead ensuring the body always stays connected with the head, and this is done through our Anchor Gaussians. Therefore, our method does not deteriorate model robustness under novel and large head rotations, which is validated in our evaluations in Table. 1, Fig. 4, and Fig. 7, which include many unseen test frames with large head rotations.

---

> > > > ### Comment · Reviewer_Q3T9 · 2024-11-20
> > > >
> > > > Thanks for your prompt update with the novel view synthesis results and further explanations. Given the additional demo videos, the model is robust under novel camera translation. For larger camera rotation, the facial area present good geometry and the connection between shoulder and head regions looks stable. In addition, there is no temporal jittering problems for shoulder area under this setting, commonly seen for monocular settings. This addresses my first concern during the last discussion.

---

> > > > > ### Author Response · Authors · 2024-11-25
> > > > >
> > > > > Dear Reviewer Q3T9,
> > > > >
> > > > > Thank you so much for your prompt update on the rating. We have just updated the paper to include additional comparison with Real3D-Portrait trained on single identity video for a fair comparison. Results and details can be found in Sec B.6, Fig. 19, and Table. 8.
> > > > >
> > > > > For Real3D-Portrait, we trained the motion adapter for 100,000 steps on a single A100 GPU, which takes around
> > > > > 80 hours. We then trained the HTB-SR model for 80,000 steps, which takes around 30 hours. Note
> > > > > that our method only requires less than 3 hours to train. From the results, it can be seen that our
> > > > > method is able to generate the head and cloth with much better quality.
> > > > >
> > > > > In Real3D-Portrait, a torso model is used to predict 2D warping from body keypoints to deform the latent image for fused body
> > > > > generation. We suppose that, while this approach can somehow learn to correctly connect the head to the body,
> > > > > without 3D-2D constraints from anchor Gaussian, it fails to learn sharp textures on the clothes. This
> > > > > result also matches our No Anchor Loss ablation in Fig 10, where the use of Anchor Loss can effectively improve the sharpness and accuracy of the cloth texture learned.
> > > > >
> > > > > We hope the updated comparison with Real3D-Portrait can further address your concerns and show the effectiveness of our novel approach.

---

> > > > > > ### Comment · Reviewer_Q3T9 · 2024-11-25
> > > > > >
> > > > > > Thanks for your additional experiment and further comparison with image-warping based transformation for shoulder rendering. From Tab. 8 in the revised Appendix, I can clearly see the performance difference even given long fine-tuning time of Real3D-Portrait. This result can demonstrate the effectiveness of using anchor gaussian + neural texture as proposed by the authors.
> > > > > >
> > > > > > From a perspective extending beyond this work, the rendering of shoulder together with head is a problem that resolves the rendering of undefined geometry from flame or other head template model. The authors have demonstrated the effectiveness of the proposed method. I think this method might be further applicable or bring insight to other similar problems (like rendering cloths beyond human body skeletons for full-body 3D-photo-real avatar generation, etc).
> > > > > >
> > > > > > Back to the topic, in summary, due to the training efficiency, performance improvement, rendering speed and also light-weight network usage (For Real3D-Portrait, there seems to be complicated CNNs and transformers for both image-warping and head-torso blending), the additional experiments resolve my concern of effectiveness of this method compared with other SOTA head+torso rendering models and I raise the final score to 8.

---

### Official Review · Reviewer_bZ83 · 2024-10-30

**Soundness:** 4
**Presentation:** 3
**Contribution:** 3
**Rating:** 6
**Confidence:** 3

**Summary:**

The paper addresses the challenge of creating realistic upper body avatars from monocular videos. Existing methods focus on head avatars and struggle with detailed body textures, resulting in blurry reconstructions. The proposed solution uses neural textures guided by sparse Gaussians to improve fidelity and speed, achieving up to 130 FPS, and excels in both self and cross-reenactment tasks.
Contributions:
1. It uses anchor Gaussians to effectively model detailed upper body textures.
2. It eliminates MLP queries for faster speeds, reaching about 130 FPS.

**Strengths:**

1. This method effectively models detailed upper body textures.
2. This paper combines 3DGS and neural fields to construct head and body textures, showcasing technical novelty.
3. Experimental results demonstrate that this method has significant advantages in rendering accuracy and inference speed while using a smaller number of 3D Gaussians.

**Weaknesses:**

1. The article lacks some clarity in its writing. For example, it does not explain what texture space is in the introduction.
2. The article could include additional baselines, such as Gaussian Avatars and Gaussian Head Avatar.
3. Regarding the rendering formula in Section 3.5, the coefficient for the third term, body texture, is the product of the opacities of the first two terms. Wouldn't this cause most of the pixels' colors in the third term to be overridden?

**Questions:**

1. FlashAvatar is motivated by inference acceleration. Has this paper calculated its FPS?

---

> ### Author Response · Authors · 2024-11-19
>
> ## 1. Texture Space
>
> We have modified the paper line 085-086 to include a more information description of the texture space we used in our method.
>
> ## 2. Additional Baselines
>
> We would like to note that both of the methods suggested are specifically multiview reconstruction methods, and they have already been cited and discussed briefly in the initial version of the paper. We have added a comparison with GaussianAvatars in Sec B.6, Fig. 17, and Table. 7 and our method achieves superior performance in both full and head-only reconstructions. Gaussian Head Avatar is not comparable to our method because it requires multiview videos to reconstruct an accurate initial neural mesh, whereas our method works only with a monocular camera. Training Gaussian Head Avatar with monocular video simply produces degraded results.
>
> ## 3. Occlusion of Body Texture
>
> Yes, body textures are rendered last and can be occluded by either the Anchor Gaussians or the head Gaussians. However, the Anchor Gaussians are sparse and have low opacities, and head Gaussians are constrained to stay only around the head part via semantic segmentation. Therefore, the body of the avatar is not occluded by either of the two and the body texture can be properly rendered and optimized.
>
> ## 4. Flash Avatar FPS
>
> We have updated Table. 2 to include the FPS of FlashAvatar. Note that since the authors did not release FPS benchmarking code, we modified the test rendering script to exclude dataloading, Gaussian extraction as well as image logging to compute the FPS. On our datasets with body and shoulders visible, FlashAvatar renders at around 130FPS, which is comparable to the FPS of ours (No MLP). Please also note that FlashAvatar is rendering with a worse quality and the number of Gaussians used is much less.

---

> > ### Comment · Reviewer_bZ83 · 2024-11-23
> >
> > Thank you for your response and the additional experiments.
> > I believe my confusion has been resolved.

---

### Official Review · Reviewer_Dd1H · 2024-11-03

**Soundness:** 2
**Presentation:** 3
**Contribution:** 2
**Rating:** 5
**Confidence:** 4

**Summary:**

This paper introduces a method to create a high-fidelity head avatar.
This method maps intricate texture to the image plane through a sparse set of anchor Gaussians driven by 3DMM, not only reconstructs the head, but also the upper body parts.
In addition, this paper also proposes a method to use Gaussian spraying to speed up rendering.

**Strengths:**

- This paper uses neural textures to model body parts simultaneously, instead of just the head.
- The paper proposes a method to accelerate reasoning, so that the rendering speed can reach 130 frames per second.

**Weaknesses:**

- This paper lacks comparisons with some methods in terms of head reconstruction quality, including [1, 2].
- The head part of this method is similar to the theory of existing methods (PSAvatar).
- The tracking results given in Figure 11 are not convincing. In many body tracking projects, some scenes containing half of the body can still produce reasonable results. For example, One-Stage 3D Whole-Body Mesh Recovery with Component Aware Transformer. The paper should describe in more detail the methods used for tracking and why they were failed, otherwise I don't see the point why not using the upper body SMPL model.

- Reference

        [1] Gaussian Head Avatar: Ultra High-fidelity Head Avatar via Dynamic Gaussians
        [2] GaussianAvatars: Photorealistic Head Avatars with Rigged 3D Gaussians

**Questions:**

- Since the modeling of the body parts starts from a canonical frame/space, will the selection of this canonical frame significantly affect the reconstruction quality?
- I still have doubts about how this method drives the reconstruction of the upper body of the character for unseen self-driving frames and cross-driving frames. From the visual results, it seems that the upper body is not aligned with the driving frame.

---

> ### Author Response · Authors · 2024-11-19
>
> ## 1. Additional Comparisons
>
> We would like to note that both of the methods suggested are specifically multiview reconstruction methods, and they have already been cited and discussed briefly in the initial version of the paper. We added additional comparisons with “GaussianAvatars: Photorealistic Head Avatars with Rigged 3D Gaussians” in Sec B.6, Fig. 17 and Table. 7 and our method achieves superior performance in both full and head-only reconstructions. “Gaussian Head Avatar: Ultra High-fidelity Head Avatar via Dynamic” is not comparable to our method because it requires multiview videos to reconstruct an accurate initial neural mesh, whereas our method works only with a monocular camera. Training Gaussian Head Avatar with monocular video simply produces degraded results.
>
> ## 2. Head Part of the Method
>
> Our main novelty and contribution lies in identifying the limitation of using Gaussian Splatting to model body and cloth with high-frequency texture, mitigating the issue with 2D neural texture warping, constraining the body texture with the use of Anchor Gaussians, as well as a homography inference approach for real-time rendering. We believe the importance of modeling high-fidelity body clothing for real-world applications is non-trivial and should not be overlooked.
>
> ## 3. Haf-Body Tracking
>
> We followed the official preprocessing pipeline of “GaussianAvatar: Towards Realistic Human Avatar Modeling from a Single Video via Animatable 3D Gaussians” to obtain the results in Fig. 11, where a landmark optimization pipeline is used to obtain the SMPL parameters. We included some SMPLX estimation results using “One-Stage 3D Whole-Body Mesh Recovery with Component Aware Transformer” in Fig.14 in the paper revision. As it is still mainly trained and optimized on frames with full-body or upper-body portraits with arms visible, under our tight framing setting, it tends to struggle with shoulders and can fail to detect any body with extreme poses such as the one shown in the last column.
>
> We believe RDd1H has potentially misunderstood our method and contribution. The main focus of the proposed method is to mitigate the issue caused by using Gaussian Splatting when fitting high-frequency cloth textures, rather than realizing shoulder movements modeling, which has already been achieved by several existing works such as GSAvtar. Even if highly accurate SMPL/SMPLX parameters are given, modeling everything using Gaussian Splatting only would still cause artifacts in high-frequency clothes, and our neural texture with Anchor Gaussian has to be adapted for better performance.
>
> ## 4. Selection of Canonical Frame
>
> Canonical frame can be chosen as any frame with an unoccluded body and natural pose and has little impact on performance. In fact, we only manually selected the canonical frame for subject 003 because she was not centered at the beginning of the video. For the rest of the subjects, we simply choose the first training frame as the canonical frame.
>
> ## 5. Driving of Upper-Body
>
> To make a faithful comparison with our baseline methods which do not support reenactment in the upper body, as they do not have bones for body and shoulders, we did not drive the upper body motion, but only used our Anchor Gaussians to ensure the body and the head stay connected.
>
> Based on the comments we received, we believe Reviewer Dd1H has several potential misunderstandings of our method and contribution. We hope our answers can clarify the misunderstanding and address the concerns raised. We would highly appreciate active discussion from Reviewer Dd1H to further clarify any questions raised.

---

> > ### Author Response · Authors · 2024-11-25
> >
> > Dear Reviewer Dd1H,
> >
> > We hope our initial response has addressed some of your concerns and clarified any misunderstandings regarding our representation and methodology.
> >
> > As the discussion period is drawing to a close, we would greatly appreciate your further feedback. Particularly, because you gave the lowest rating among all reviewers, we believe it would be particularly helpful, for us and the ACs, to further understand the reasoning behind your evaluation, as well as your thoughts after reviewing our response.
> >
> > We look forward to your comments and will provide additional clarifications for any additional questions raised.

---

> > > ### Comment · Reviewer_Dd1H · 2024-11-25
> > >
> > > Thank you for your reply and additional experiments. My doubts have been generally resolved and I decided to raise my rating to respect the discussion. But I still worry that the contribution is incomplete due to the lack of shoulder control and only improving the rendering details.

---

> > > > ### Author Response · Authors · 2024-11-28
> > > >
> > > > Thank you for your thoughtful feedback and for raising your score following our responses and additional experiments. We appreciate your acknowledgment of the improvements shown in the method.
> > > > Regarding your concern about modeling of the shoulder motion, we would like to further clarify the following:
> > > >
> > > > ## Shoulder Movements
> > > > While our primary goal was to address the limitations of Gaussian Splatting for high-frequency texture modeling, we do understand the importance of modeling detailed shoulder dynamics such as shrugging. As you noted, incorporating full body tracking and adding explicit shoulder control through SMPL bones would indeed complement our method. However, we deliberately chose not to introduce such components to remain consistent with the baselines used in our comparisons, which also lack explicit shoulder bones. We believe that incorporating this in future work could make the method even more comprehensive.
> > > >
> > > > ## Rendering Details as a Contribution
> > > > Improving rendering details is a critical contribution, especially for practical applications where high visual fidelity is essential such as video conferencing. The proposed Anchor Gaussian-constrained neural texture warping provides a novel approach to overcoming the inherent limitations of Gaussian Splatting in modeling textures for complex surfaces like clothing. We believe this advancement is non-trivial and addresses a significant gap in existing avatar reconstruction methods. Also, compared with baselines used in our comparisons and other SOTA head+torso neural rendering model including Real3D-Portrait and StyleAvatar, our method is superior in training efficiency, performance improvement, rendering speed, and light-weight network usage.
> > > >
> > > > ## Broader Impact and Potential
> > > > Beyond its immediate contributions, we believe this work lays the foundation for tackling broader challenges in avatar rendering and reconstruction. As highlighted by Reviewer Q3T9, the integration of head and shoulder rendering effectively addresses the issue of undefined geometry in FLAME and similar head template models. This approach not only demonstrates the effectiveness of Anchor Gaussian-constrained neural texture warping but also opens doors for further exploration. We hope it could further inspire advancements in rendering clothing beyond human skeletal models, enabling full-body photorealistic 3D avatar generation for applications in entertainment, virtual reality, and telepresence. By addressing high-frequency texture limitations and proposing efficient solutions, we believe this method has the potential to influence both academic research and practical developments in the field of 3D generative modeling.

---

### Official Review · Reviewer_CZUM · 2024-11-04

**Soundness:** 3
**Presentation:** 1
**Contribution:** 2
**Rating:** 6
**Confidence:** 3

**Summary:**

This paper proposes a Gaussian splatting based deformable avatar method with a specific focus on modelling shoulders with a high fidelity. The head is represented by a set of 3D Gaussians while the shoulders are represented by a neural texture map. The gaussians on the head are deformed from the canonical space using Linear Blend Skinning while the texture-map for the shoulders are warped using a pose dependent MLP. Once in the deformed space, both the neural textures and the gaussians are rendered to yield the final image. The properties of the gaussians and the warpings are learnt through a set of RGB and deformation based losses. The neural texture warping is specifically constrained through an anchor loss defined using a set of Gaussians learnt on the shoulders. Both quantitative and qualitative show improvements over prior work.

**Strengths:**

1) The paper is well written and easy to follow
2) The use of a neural texture map to model the shoulders is well motivated and intuitive
3) The additional loss terms introduced are well ablated in both the main paper and the supplementary
4) The proposed method both qualitatively and quantitatively outperforms prior work

**Weaknesses:**

1) There are no significant out-of-plane view synthesis results that are shown in the results to demonstrate whether or not the learn texture map is able to generate novel renders of the shoulders in a geometrically consistent way or it just renders them as a plane

2) This paper is, I believe, a little out of scope of ICLR and is more suited at a vision or graphics conference. However, I will defer to other reviewers and the AC for that before I consider it as a strong weakness.

**Questions:**

1) As mentioned above, I’m curious to see novel out-of-plane renders of the shoulders themselves to see how geometrically accurate they are.

2) There is no ablation of exactly how robust the proposed method is to shoulder movement in the capture video. Understanding this would help evaluate the method more throughly (I do grant that prior work does not even address this problem, but I believe this is important to make the paper more complete.)

3) Any reason why Splatting-avatar results are so much worse compared to the other methods?

---

> ### Author Response · Authors · 2024-11-19
>
> ## 1. Out-of-Plane Results
>
> We show novel view synthesis results in Sec B.5 and Fig. 15 in the revised version of the paper. Please note that novel view rendering with extrapolated views is challenging for all neural avatar methods from monocular video. Specifically, methods such as StyleAvatar which models the body in 2D cannot render from novel views at all. In comparison, our use of Anchor Gaussians as a 3D-2D constraint allows us to render novel views that are close to training views while ensuring connectivity between head and body.
>
> ## 2. Robustness to Shoulder Movements
>
> Our method shows strong robustness to various large body movements including both rotation and translation, as shown in Fig. 3. However, please note that to keep a fair comparison to the baseline methods which only use head 3DMM FLAME without any body bones, we similarly only warp the body anchor Gaussians with no bones for shoulders or chest. Therefore, modeling of specific shoulder movements such as shrugging is not considered in our evaluation.
>
> ## 3. Performance of Splatting-Avatar
>
> We followed the official instructions and guidelines from the authors to compare with Splatting-Avatart. We also sought advice from the authors to tune the hyperparameters, but failed to obtain improved results. Therefore, the results included in the paper are all obtained by following the official settings.

---

> > ### Author Response · Authors · 2024-11-20
> >
> > As suggested by Reviewer Q3T9, we have updated the supplementary materials to include additional novel view synthesis videos. We included results with larger camera rotation as `supplementary/novel_rot.mp4`, as well as large novel camera translation results as `supplementary/novel_trans_x.mp4`, and `supplementary/novel_trans_y.mp4`. We recommend Reviewer CZUM to also check those results as a question regarding out-of-plane view synthesis was raised.

---

> > > ### Author Response · Authors · 2024-11-25
> > >
> > > Dear Reviewer CZUM,
> > >
> > > We hope our initial response has resolved some of your concerns and clarified potential misunderstandings regarding our representation and method.
> > >
> > > Since the discussion period will end soon, we would highly appreciate your feedback and further discussion. We highly recommend you to view the amended video results with larger camera rotation in `supplementary/novel_rot.mp4`, as well as large novel camera translation results `supplementary/novel_trans_x.mp4`, and `supplementary/novel_trans_y.mp4`. We would like to highlight that Reviewer Q3T9, who has concerns similar to yours in novel view synthesis and large shoulder motions, has updated the rating from 5 to 6 after checking the amended videos.
> > >
> > > We look forward to your comments and will further clarify any questions raised.

---

> > > > ### Comment · Reviewer_CZUM · 2024-11-25
> > > >
> > > > I would like to thank the authors for their response. While the new results do demonstrate the shoulders seem to modeled incorrectly as a flat plane, the renders themselves are not too terrible considering the limited training views of the shoulders. Looking at the other reviews and the rebuttal, I have decided to raise my score. Though, I must mention that I am still unsure about how well this paper fits in the scope of ICLR.

---

> > > > > ### Author Response · Authors · 2024-11-28
> > > > >
> > > > > Thank you for your thoughtful feedback and for raising your score following our responses and additional experiments. We appreciate your acknowledgment of the improvements shown in the method.
> > > > >
> > > > > Regarding the observation of shoulders being modeled as a flat plane, this arises from the constraints of monocular video data, where sufficient variations in shoulder visibility are not present. This limitation is shared among all per-identity avatar modeling methods using monocular views. Despite this, we focused on ensuring **stable and coherent** head-body connections and achieving high rendering fidelity in scenarios with limited training data. Expanding the method to better handle complex 3D shoulder geometry under novel views is an important avenue for future work.
> > > > >
> > > > > As for your concern regarding the paper’s fit within the scope of ICLR, we would like to highlight that Real3D-Portrait , suggested by Reviewer Q3T9 for comparison, is accepted by ICLR 2024. This demonstrates that ICLR welcomes contributions advancing neural rendering and generative models applied to avatar reconstruction tasks. Our work builds on this direction, introducing the novel concept of Anchor Gaussian-constrained neural texture warping to address specific challenges in modeling high-frequency textures and ensuring robust head-body connections. We believe our method complements such efforts and contributes meaningfully to the community.

---

### Official Review · Reviewer_SS61 · 2024-11-04

**Soundness:** 3
**Presentation:** 3
**Contribution:** 3
**Rating:** 6
**Confidence:** 4

**Summary:**

This paper proposes a method to reconstruct upper body avatars from videos based on 3D Gaussian Splatting. The method models head and shoulders separately, i.e., heads by 3D Gaussians anchored on 3DMM template, and shoulders by Anchor Gaussians and neural textures. The paper proposes a Neural Texture strategy to model shoulders with less 3D Gaussians. The authors evaluate their methods on different experimental datasets and human subjects.

**Strengths:**

The paper has a clear structure, and it‘s easy to follow.
The idea of modeling the head and shoulder separately is interesting. The neural texture strategy can model the shoulders efficiently with less 3D Gaussians.
The pipeline is verified on different human subjects.

**Weaknesses:**

The method models head and shoulders separately, i.e., different Gaussian representations and different tracking approaches. This causes a limitation, i.e., the head and shoulders are misaligned for some relatively big body movements.

It seems that the Anchor Gaussian only supports relatively simple should motions. It’s not verified that the proposed method can model pose-dependent wrinkles for the shoulders in the paper. The body movements in the supplementary demo are too simple, and I cannot observe that the clothing wrinkles vary with poses.

**Questions:**

Considering the 3DMM parameters and landmarks of the shoulder are acquired separately, is there a method to unify the two tracking results? It seems that the head and shoulder are misaligned in some frames.

---

> ### Author Response · Authors · 2024-11-19
>
> ## 1. Separated Tracking for Head and Body
>
> There are several methods that can achieve unified tracking of both head and body through SMPLX parameter estimation, as shown in Fig.14 in the paper revision.  However, as they are still mainly trained and optimized on frames with full-body or upper-body portraits with arms visible, their performance can be degraded with our tight framing setting (video conference etc.), where most of the lower arms are not visible: they tend to struggle with shoulders and can fail to detect any body with extreme poses such as the one shown in last column. The misalignment issue is likely to be further improved with more robust and accurate unified tracking. Regardless, we found that the separated tracking works fine when modeling general talking videos without exaggerated body movements, as suggested by our leading performances in the experiments.
>
> ## 2. Pose-Dependent Wrinkles
>
> We have added visualization of pose-dependent appearance changes for test frames in Fig. 13. Fig. 14 also shows strong pose-dependent appearance changes during novel view rendering. Please note that pose-dependent fine texture is mainly used to deal with wrinkles and lighting changes as noises in the training frames, modeling them exactly the same as the ground truth at test time remains a challenging problem.

---

> > ### Comment · Reviewer_SS61 · 2024-11-26
> >
> > I would like to thank the authors for their response. The main challenge in this problem is the tracking of both head and body. This paper did not solve the tracking problem, and instead tracks the head and body separately, which causes the misalignment problem. In addition, the solution for body part can only handle some simple motions. However, the proposed method still outperforms the existing baselines, and the performance is acceptable. Therefore, I maintain my original score.

---

> > > ### Author Response · Authors · 2024-11-28
> > >
> > > Thank you for acknowledging the rebuttal and updated results. Regarding your concern about modeling of shoulder, we would like to further clarify the following:
> > >
> > > ## Shoulder Movements and Tracking:
> > > We would like to note that improving the body tracking quality and consistency is not within the scope of this paper, and we do not claim any contribution related to it. Our primary goal was instead to address the limitations of Gaussian Splatting for high-frequency cloth texture modeling.
> > >
> > > We do acknowledge the importance of modeling detailed shoulder dynamics (e.g., shrugging), which will further enhance the application of the method. We believe that incorporating this in future work could make the method even more comprehensive. However, we deliberately chose not to introduce such components to remain consistent with the baselines used in our comparisons, which also lack explicit shoulder bones and shoulder modeling.
> > >
> > > ## Broader Impact and Potential:
> > > Beyond its immediate contributions, we hope this work lays the foundation for tackling broader challenges in avatar rendering and reconstruction. As highlighted by Reviewer Q3T9, the integration of head and shoulder rendering effectively addresses the issue of undefined geometry in FLAME and similar head template models. This approach not only demonstrates the effectiveness of Anchor Gaussian-constrained neural texture warping but also opens doors for further exploration. We hope it could inspire advancements in rendering clothing beyond human skeletal models, enabling full-body photorealistic 3D avatar generation for applications in entertainment, virtual reality, and telepresence.

---

### Author Response · Authors · 2024-11-19

We would like to thank all reviewers for their invaluable comments and feedback. We have replied to each reviewer individually to address their questions and concerns, and have updated the paper to include extra results and figures. We would like to highlight and answer some concerns raised in common:

## 1. Body 3DMM Tracking

We have included some results of SMPLX 3DMM estimation via one of the latest methods suggested by Reviewer CZUM, ”One-Stage 3D Whole-Body Mesh Recovery with Component Aware Transformer”, in Fig. 14. It shows improved performance compared to landmarks optimization method, but still predicts incorrect shoulder position, misaligned body rotation, and can fail to detect any body under challenging poses. This is expected as detecting exact body pose when only shoulders are visible is highly challenging due to various body shapes and clothing, and also because existing SMPLX detectors are typically trained and optimized on images with a wider framing.
We wish to clarify that, the main focus of the proposed method is to mitigate the issue caused by using Gaussian Splatting when fitting high-frequency cloth textures, rather than realizing shoulder movements modeling, which has already been achieved by several existing works such as GSAvtar. We decided to use only head 3DMM (FLAME) as the backbone 3D representation for LBS because 1) it allows a fair comparison with other baselines with strong performance in fine texture representation, as they only use FLAME; 2) the existing SMPLX prediction methods still lack accuracy and reliability in predicting correct body position and rotation. We believe our proposed method and experiments can effectively exploit the issue of fitting high-frequency texture with Gaussians and the benefits of replacing it with Anchor Gaussian-constrained texture wrapping.

## 2. Shoulder Movements Modeling

As described in the previous paragraph, the main focus of the proposed method is to mitigate the issue caused by using Gaussian Splatting when fitting high-frequency cloth textures, rather than realizing shoulder movements modeling, which has already been achieved by several existing works such as GSAvtar. Therefore, our method and all baselines do not use any bones for shoulders that allow the modeling of specific shoulder movements such as shrugging. However, we can certainly model large body movements such as translation and rotation, as shown in Fig. 3, as well as non-rigid transformations in the clothes.
Additional Comparisons: As requested, we include additional comparisons with GaussianAvatar, and Real3D-Portrait in Sec B.6, Fig. 17, and Table. 7. Our method achieves superior results in both full and head reconstructions. We did not compare with GaussianHeadAvatar, because it is a multiview reconstruction method that requires accurate neural mesh to be reconstructed as an initialization. In contrast, our method uses only a monocular view. We included results for two subjects due to constraints in computational resources, and will include a comparison of all subjects in the camera-ready version if accepted.

We hope our answers and updated paper can help to address the concerns raised in the initial reviews. Since we received highly mixed comments from the reviewers, we would highly appreciate active discussions from the reviewers and we are happy to clarify any further questions.

---

### Meta-Review · Area_Chair_pEAK · 2024-12-20

**Metareview:**

This paper proposes a method for constructing a head and shoulders avatar from a provided video of a subject. The main innovation proposed in the approach is to employ regular Gaussian splatting for the face along with sparse anchor Gaussians and a view-dependent neural texture for the relatively simpler shaped shoulder region, which normally has more complex texture from clothing. The latter approach is motivated by the fact that highly textured regions require many Gaussians to model the texture making the approach unnecessarily computationally expense, which can be saved by employing warping of neural textures instead. The authors show state of the art results both in terms of accuracy and speed versus the existing state of the art.

The work address a relatively un-addressed problem in 3D human avatar creation of including the shoulders as well. The majority of the existing works constrain themselves to modeling the face region only. In most practical applications e.g., video conferencing, however, the lack of shoulders, severely limits realism and hence the end user experience.

On the flip side, the major limitation of this technique is its dependence on two separate approaches for tracking the face and shoulder movements, which leads to inconsistent seams between them. This can be partially addressed by using SMPL-X-based full body tracking methods like OSX, but the proposed method still is limited in only being able to accurately model small shoulder motions like rotations and translations and not larger ones like shrugging. This limits the novelty of the method in mainly proposing a novel way to efficiently handle the modeling highly textured regions of the human body only.

**Additional Comments On Reviewer Discussion:**

Five reviewers provided the final scores of 6, 6, 5, 6, 8. The reviewers expressed concerns around inconsistent face and body tracking, limited shoulder movement modeling, lack of comparisons to various state-of-the-art baselines and the lack on novel view results, among others. While many of the reviewers' concerns were addresses in the rebuttal phase resulting in several reviewers increasing their final rating mostly towards marginal acceptance, the fundamental limitations inherited by the proposed method by split body tracking and limited shoulder movements remained unaddressed.

Overall, the AC feels that this work's contribution of efficiently modeling highly textures clothing via neural texture and anchor Gaussians,, which could benefit and inform the design of whole-body avatars in the future;  along with its improved quantitative and qualitative results compared to the state of the art places it slightly above the bar for acceptance.

---

### Decision · Program_Chairs · 2025-01-22

Accept (Poster)